# Metastable Coordination Dynamics of Collaborative Creativity in Educational Settings

**Carlota Torrents [1,\*], Natàlia Balagué [2], Robert Hristovski [3], Maricarmen Almarcha [2] and J. A. Scott Kelso [4,5]**

[1] Complex Systems in Sport Research Group, National Institute of Physical Education of Catalonia (INEFC), University of Lleida, 25192 Lleida, Spain

[2] Complex Systems in Sport Research Group, National Institute of Physical Education of Catalonia (INEFC), University of Barcelona, 08038 Barcelona, Spain; nbalague@gencat.cat (N.B.); mcalmarcha@gencat.cat (M.A.)

[3] Complex Systems in Sport Research Group, Faculty of Physical Education, Sport and Health, Ss. Cyril and Methodius University, 1000 Skopje, Macedonia; robert_hristovski@yahoo.com

[4] Human Brain and Behavior Laboratory, Center for Complex Systems and Brain Sciences, Florida Atlantic University, Boca Raton, Fl 33432, USA; jkelso@fau.edu

[5] Intelligent Systems Research Centre, University of Ulster, Derry~Londonderry BT48 7JL, UK

\* Correspondence: ctorrentsm@gencat.cat; Tel.: +34-973272022

**Abstract:** Educational systems consider fostering creativity and cooperation as two essential aims to nurture future sustainable citizens. The cooperative learning approach proposes different pedagogical strategies for developing creativity in students. In this paper, we conceptualize collaborative creativity under the framework of coordination dynamics and, specifically, we base it on the formation of spontaneous multiscale synergies emerging in complex living systems when interacting with cooperative/competitive environments. This conception of educational agents (students, teachers, institutions) changes the understanding of the teaching/learning process and the traditional roles assigned to each agent. Under such an understanding, the design and co-design of challenging and meaningful learning environments is a key aspect to promote the spontaneous emergence of multiscale functional synergies and teams (of students, students and teachers, teachers, institutions, etc.). According to coordination dynamics, cooperative and competitive processes (within and between systems and their environments) are seen not as opposites but as complementary pairs, needed to develop collaborative creativity and increase the functional diversity potential of teams. Adequate manipulation of environmental and personal constraints, nested in different level and time scales, and the knowledge of their critical (tipping) points are key aspects for an adequate design of learning environments to develop synergistic creativity.

**Keywords:** coordination dynamics; cooperative learning; synergy; constraints; collaborative creativity; complex systems; competition; self-organization; education

## 1. Introduction

Fostering creativity is one of the essential aims of education in the 21 century [1]. In an incredibly fast changing world, to be creative has become one of the most valued traits of personality and, nowadays, creativity is essential to deal with the challenges of sustainable development [2]. It is not necessary to be a "genius", or to be the best in a specific domain (big C-creativity) in order to be creative, as creativity is essential to cope with our daily lives, the so-called little-c creativity [3]. Writing a post, coming up with a funny joke, inventing a tale for your daughter, designing nice teaching material or finding new ways to foster knowledge and competences related with sustainability are some examples of c-creativity.

Collaboration among persons and the configuration of teams with diverse expertise seem crucial to solve the nested social, scientific or technical problems of our society, and especially those related with sustainable development [4] to encourage individuals to

find new directions for active participation and societal cooperation [5,6]. Education is the more powerful means to foster critical thinking, develop sustainability values and prepare individuals for working cooperatively to achieve common goals.

Both terms, cooperative learning (CL) and collaborative learning are used to define processes that increase sustainable development competencies [6]. Although the first one is more common in the education literature, the second one also refers to instructional arrangements that involve two or more students working together on a shared learning goal. It is sometimes differentiated from CL because it does not specify the division of labor [7]. In the creativity literature, it is more frequent to use the term collaborative creativity, that refers to processes of creation based on individuals working together [8]. Scientists in this field consider that each goal structure has its place [9], but it is usual in education to "use" trendy methods or pedagogical strategies instead of being sensitive to each specific context, goal and situation. Understanding the general principles that rule the interactions among individuals is the first step before deciding on a trustable and effective method to be used in education. For this reason, understanding the processes that foster creativity in individuals and teams, as well as how cooperation emerges among them, is crucial to propose effective collaborative creativity pedagogical strategies.

Learning to collaborate does not mean teaching students the best way to cooperate, and cooperation is not necessarily linked to creativity. Accordingly, collaborative creativity is not something to be imposed externally by a teacher or educational system. These kinds of imposition are based on the assumption that someone out of the individuals or team knows which is the best solution in each specific situation, or the steps that team members should follow to reach the right solution (assuming that someone knows in advance which it is). These assumptions ignore the properties of complex living systems (students, teachers, teams).

In this paper we propose to focus on understanding and studying the coordination dynamics (CD) principles allowing the emergence of collaborative creativity in teams (of students, teachers, students and teachers or educational institutions). For CD, collaborative creativity supposes a subtle blend of cooperative and competitive processes. Cooperation, for example, is needed to compete with challenging environments. Under such a framework, the roles of all agents are substantially transformed. Instead of establishing the right solutions and proposing sequences of actions, teachers co-design with students challenging and meaningful learning environments that stimulate competition and cooperation. Students cooperate to explore, with the help (or not) of the teacher, creative ways to solve the problems with increasing efficacy. The student–teacher relationship is that of a dynamic, complementary pair [10,11]. The dynamic nature of the process offers infinite possibilities; however, they are limited by the constraint of sharing common goals and values. In this sense, the age difference between teachers and students might be a limitation to which teachers should be sensitive. Gender stereotypes and the different behavior observed in men and women [12,13], should be also taken into account when applying collaborative strategies to guarantee equitability.

A survey of the literature reveals that the key to adaptation in teams is the dynamic nature of the coordination between members or the formation of synergies [14,15]. We will have more to say about this in the next section. For now, we note that synergies are not based on rigid coordinative states: they possess both cooperating and competing aspects, and with all the members working in coherence [11]. In social settings, this coherence seems to be achieved by sharing meaningful goals and allowing the emergence of individually diverse behavior. The examples described in this paper refer mostly to student teams or student–teacher teams. Nevertheless, the same principles can be used to understand the formation of synergies between teachers and schools or even between education institutions. In addition, we claim that these principles are valid for any social context, not only educational ones.

Current creativity research has been mainly focused on studying how new ideas come to the mind of individual agents and finding ways to promote the process. Foster-

ing a creative mind is one of the most popular aims of creativity research, and thus, the most used tests to measure creativity are based on counting the number and variety of ideas that individuals can produce in a specific task. The best personal traits related to creative states have been described, such as the best mood state [16,17] or the impact of rewards [18]. Neuroscience has also studied the neural correlates of creative cognition, suggesting that creativity involves a complex interplay between spontaneous and controlled thinking, as well as flexible reconfigurations of dynamic functional connectivity [19]

The problems of traditional approaches to explain creativity are especially revealed when studying improvisation-based activities and/or when the focus is on the team and not on the individual. The serial view of the "perceive-think-act" model of cognition has evolved to include perception-action coupling [20–22]. The universal need for fully-fledged plan representations becomes questionable, while the role of the environment to create collective dynamics and larger functional structures are considered. Movement is inherently creative. Consider bite-block speech for example. The speaker has never encountered a bite block before, but can still produce the correct vocalic sound, even when the jaw is fixed and related structures are anesthetized [23].

Dynamic approaches have been recently used to conceptualize creative behavior and to understand the mechanisms involved during the creative process [24–27]. They consider creativity as a changing process in time, in contrast to the static assumptions of more traditional approaches.

The aim of this paper is to discuss how synergies emerge spontaneously when students share meaningful goals and how these synergies, embracing processes of both cooperation and competition, increase their functional diversity (the system becomes more diverse but also more functionally uncertain for a given environment, [28]) and creativity (see Figure 1).

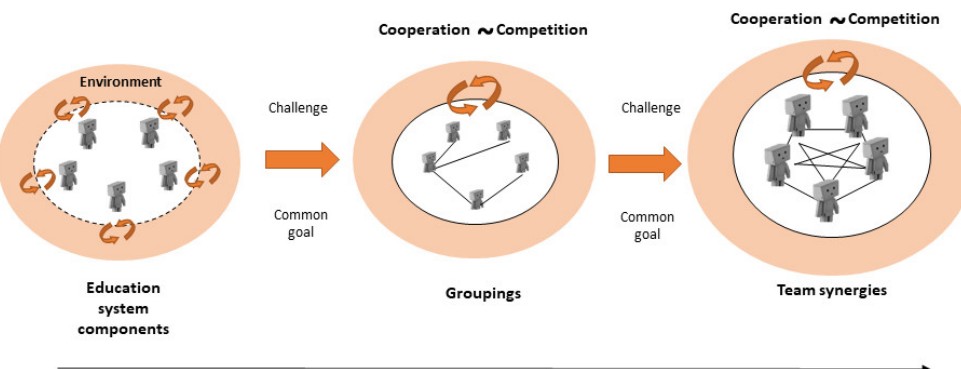

**Figure 1.** Development of collaborative creativity in groups when education system components are engaged in metastable cooperation–competition processes. Left circle: education system components interacting individually with the environment. Middle circle: individual components, facing a meaningful environmental challenge and sharing common goals, reduce their individual

behavior to favor grouping and cooperation. Right circle: team synergies emerge and develop when groupings face enough relevant environmental challenges. In such teams, not only does the whole group increase its functional diversity potential and creativity, the individual components, interacting with flexibility, also enhance their creative individual expression.

## 2. Understanding the Emergence of Synergies in Teams: Coordination Dynamics (CD)

Complexity sciences were introduced for studying the formation of teams, especially in sport [29], after the systematic research established through the emerging science of coordination, coordination dynamics (CD). CD refers to the concepts, methods and tools used to describe, explain and predict how patterns of coordination form, adapt, persist and change in living systems [30]. To coordinate with themselves, with others, or with the environment, living systems form synergies, which are functional groupings of elements that are temporarily constrained to act as a single coherent unit. Components interact to form synergies and those synergies, in turn, govern the components' behavior [31,32]. A complementary view is that the parts become differentiated via a process of intra-action within the wholistic synergy itself. That is to say, as parts, the organizm and the environment are never separate (see [33] for discussion)

The need to coordinate with others arises in early life and often in daily life without the necessity to be taught: clapping hands with one's parents, conversing or walking in synchrony with a colleague [34,35]. CD has shown that interpersonal coordination is achieved by linking the degrees of freedom into synergies. The two basic features of synergies are dimensional compression (the synergy possesses lower dimensionality than the set of components from which it arises) and reciprocal compensation (the ability of one component of a synergy to respond adaptively to changes in others) [35]. Both aspects have been observed in multiple interpersonal coordination tasks [36]. The hallmark of a synergy is that the individual components spontaneously adjust their behavior to sustain the integrity of function [37]. Natural variations are compensated by adjustments or co-variations [23]. In a synergy, different components can produce the same function (degeneracy), and the same components can be assembled to produce multiple functions (pleiotropy) [29,30,33].

Synergies are self-organized, that is, they emerge spontaneously without needing hierarchical command and control. In this way, coordination patterns arise as a consequence of the dynamics of the system with no need for specific order imposed from the outside or inside. Spontaneous self-organizing tendencies interact, guiding or directing such tendencies in specific ways [14,36,38].

Social interactions exhibit lawful coordination patterns at multiple levels of description characterized by the emergence of functional synergies (from microscopic levels such as genes, cells, and neurons, to macroscopic levels such as persons and persons with the environment). CD has examined these laws empirically, embedding the observations in mathematical models, which describe the intrinsic dynamics of the unit under study (e.g., in the classroom the unit will be the individual or the team) and how these units interact with the constraints imposed by their relation with other units (persons, teams of persons). When interacting, individualist tendencies for the diverse persons to express themselves coexist with tendencies to couple and cooperate as a whole [14,33]. Intrinsic dispositions and social influences are complementary aspects of social interaction [11,35].

One of the discoveries of CD is that when symmetry is broken, the system's dynamics are metastable: under the same context a system may stay for a long time in one behavior and then adaptively switch to another (e.g., [34,38]). Metastable behavior arises from the interplay of weak coupling (affording flexible binding) and component diversity. Recent experimental work and theoretical/computational modeling demonstrates its ubiquity at different scales [39]. Metastable CD explains the collective behavior of systems whose components mutually affect each other without being trapped or locked in a fixed pattern. In terms of social coordination and team formation, the dynamical mechanism of metastability is manifested at all scales, suggesting that the emergence of syner-

gies arises as long as individuals interact with flexibility (and not with rigid and imposed roles) and are allowed to express their individual autonomy (see Figure 1). Inside creative teams, two competing tendencies co-exist in a metastable way: the individual tendency to couple and the tendency to behave independently. Both tendencies are present during any creative process. For instance, leaders emerge spontaneously in teams and group members follow her/his ideas, but due to the dynamic nature of the process, other ideas emerge in some team members that may compete with the initial goal. Whereas in static systems only the first idea (the leader's idea) grows, in metastable systems, different (potentially conflicting) ideas may co-exist. In newly formed groups such different ideas may bring some degree of disorganization and dysfunctionality; in contrast, in consolidated teams flexible and diverse synergies are beneficial and may increase functionality.

Synergies are formed at many interacting nested levels (e.g., social, personal, physiological, cellular, genetic). In turn, all constraints acting on the system are nested and correlated among them through circular causality. The concept of constraints refers to boundary conditions, limitations that apply restrictions to the degrees of freedom of a system, thereby influencing the trajectories that the system may exhibit [40,41]. Nevertheless, due to the nested nature and relatedness of such constraints, they can also stimulate creativity, as the system releases some constraints in a compensatory manner to foster goal achievements [27]. The parts may interact to form a synergy, but once formed, the process of synergizing influences the behavior of the parts in a reciprocally causal fashion [15,42]. At an individual level, the psychological state of one member of a team, arising from the interaction of her/his personal and environmental constraints, can affect his/her interaction with another member, and affect the performance of the whole team. At an institutional level, the value given by a school to the enhancement of creativity may influence the motivation of teachers which, in turn, will influence the performance of the whole education community [40]. Slow changing constraints influence more permanently the system compared to fast-changing constraints. In this way, an intervention at the level of slow changing constraints may persist longer, and thus, be more effective. Personal values, for instance, create a long-term context impinging on other faster changing variables, such as the motivation, attention and actions made by peers [40].

Team behavior is not merely the result of the sum of individual behaviors, as through interactions among individuals, collective properties and behaviors that cannot be ascribed to any specific team member emerge. Synergies become most prominent when they are functionally advantageous. If the formation of a team is not more beneficial than the individual without the team, probably the team synergy—a synergy of synergies—will not be assembled. The recruitment and dissolution of synergies is a dynamic process, and synergies are assembled to accomplish the functional needs of individuals and the demands of the environment. When conditions become critical the synergy may become unstable and switch spontaneously to another synergy [42,43]. Thus, the sustainability of teams directly depends on its functionality, which is achieved through a continuous process of complexification (diversification and specialization of performance) [44,45]. Students, teams of students or all the students and staff of schools are complex systems [46,47], whose behavior evolve in response to multiple personal constraints (e.g., social values, experience, mood states, etc.) interacting co-adaptively with the environment. Such systems spontaneously form structural and functional couplings among components (synergies) to achieve task goals [31,47]. Synergies define the level of collaborative creativity that teams exhibit. High diversity or high originality of synergies only emerge when the environment requires it [47]. This refers to the principle of sufficing, that is, systems develop a sufficiently large potential relative to the environmental constraints, but do not develop the maximum of their diversity potential if they are not constrained to do it. As soon as the problem is solved, teams no longer explore other possible solutions.

Finally, the last principle of CD that needs to be further emphasized is that synergies do not only involve cooperative mechanisms. Synergies are often used to mean cooperation, but in CD synergies possess both cooperative and competitive aspects understood as metastable complementary tendencies [11]. The formation of new synergies involves the competition between the pre-existing repertoire of the members of the team, which in turn influences, if not determines, the team's repertoire, and the new behaviors to be achieved. In this sense, CD points to a subtle blend of cooperation and competition as essential to what matters (cooperation–competition).

*Constraining to Foster Teams' Creativity in Teaching–Learning Processes*

Teams are not part of the context in which members perform innovatively and creatively but are the innovative and creative entity targeted by learning designs [48]. Considering students and teams as complex systems, it is assumed that their behavior evolves in response to physical and informational constraints which interact non-linearly and co-adaptively [44,47].

The emergence of a constraints-led approach to movement is grounded in a set of seminal papers by Kugler, Kelso and Turvey [41,49,50], which brought to light, in particular, the work of the theoretical biologist Howard Pattee. Constraint-based rather instruction-based approaches have been applied to motor learning, control and sport using ideas from CD and ecological psychology [29,51,52]. Constraints acting on human behavior have been divided into organismic, environmental and task-related [51]. Task constraints are relational variables distributed between organismic (task goal or intention) and environmental demands [40]. If the task goal is not meaningful for a member of the team who has no intention of accomplishing it, the task does not act as a constraint for that individual. When team components do not share goals, team synergies have more difficulty emerging and vice versa.

Whereas traditional approaches to creativity consider the individual as the sole unit of analysis, the central role of synergies in CD promotes the team of persons as the main unit in collaborative creativity [53]. In CD, it is not so much that the whole is greater than the sum of its parts, but rather in a synergy the collective acts as an individual and the individual acts as a collective, that is, individual–collective is a complementary pair [11,14,31].

As student behavior cannot be understood independently from its context (the classroom, the classmates, the teacher, the school, their families, etc.), the role of the teacher is to design challenging contexts to promote the emergence of collaborative creativity. Thus, instead of playing the direction setter role, teachers create contexts in which learners are pushed to innovate.

## 3. Understanding Collaborative Creativity and Cooperative Learning (CL) from CD Perspective

Cooperation seems essential to survive in a challenging world. The development of creative products is often a consequence of the collaboration among different people working together to achieve a common goal. As mentioned above, the literature on creativity has tended to focus attention on individual cognitive processes, although more recent attention is increasingly put on collaborative creativity, particularly in the area of innovation and generation of new ideas. Many techniques have been developed to organize multiple ideas, such as brainstorming, mind maps or apps with similar proposals. Paradoxically, literature on brainstorming has revealed that sometimes groups generate fewer ideas than the same number of individuals working in isolation [54]. Considering that theoretically a team should perform better, Baruah and Paulus [55] analyzed some factors that could explain this phenomenon, including: groups who speak or respond by waiting for their turn (losing time in the process); the inability to express ideas fluently as they come to mind; evaluation apprehension, social loafing (letting others in the group do the work) and downward comparison (a convergence toward the performance level

of low performers in the group). In order to enhance the effectiveness of these methods, authors propose the use of brainwriting, hybrid brainstorming, short training sessions or to keep groups small. Nevertheless, this approach has its shortcomings. It is solely focused on the generation of ideas, as well as on a hierarchical model of the educational process in which the teacher organizes the teams and proposes the methods to be used. Principles of informationally coupled with self-organizing dynamical systems (CD)—that promote the emergence of coordinated and creative behavior in a changing environment or the creation of products—are thereby forgotten or ignored [26].

In order to find how best to assemble a group of people while making a creative product, Monechi, Pullano and Loreto [56] proposed developing 3D artworks in open-ended environments using LEGO bricks. Social interactions were registered, as well as the growth of each artwork. Not surprisingly, observations revealed that faster growth was more likely to occur when working teams had a high level of commitment and possessed specific topological features, including the presence of distinct "influencers". Such proposals are clearly related to those made in CL research. CL is defined as an educational approach in which small and heterogeneous groups work together to maximize their own and each other's learning [57]. In recent decades CL has become a well-recognized pedagogical practice to promote learning on the part of many scholars and researchers. In their influential book, Johnson and Johnson [58] consider some of the basic elements that mediate the effectiveness of CL such as expanding positive interdependence to include individual and group accountability, promoting interactions that facilitate goal accomplishment, and using social skills that facilitate group processing.

Some studies have analyzed how the CL approach can enhance creativity in educational fields, such as among scientific pre-scholars [59], boosting creativity and motivation in language learning [60], promoting creative thinking in higher education [61] or reading and writing in primary classrooms [62], among others. However, these studies compare strategies based on CL with individual-based learning, but do not compare different cooperation or collaborative approaches or treat teams as units of analysis. Team creativity is based on the idea that the resulting output has to be more functionally diverse [28], innovative as well as useful [63] or more pleasant than the outputs obtained individually—or more than the sum of the creative outputs of its individual group members. The work of each individual influences and positively motivates the work of others in the group [64].

Following principles of self-organization and the central role of synergy formation [23,31], these proposals (i.e., collaborative creativity) could be improved by including competitive processes when describing team functioning, as well as considering causality and the nestedness of constraints. Traditionally, competitive learning means that students look for outcomes beneficial for themselves but detrimental for others [9]. However, competition is also referred to processes defending one's own actions or decisions, opposed to others, but beneficial for all. Also, competitive processes may be related to overcome environments incompatible or in the opposite direction to the task goal.

CL was a reasonable and praiseworthy advancement when competition dominated educational methodologies, following behaviorism and an emphasis on individualistic and programmed learning. In our opinion, these methodologies can be improved by recognizing the significance of synergies and promoting—meaning creating the conditions for—their spontaneous emergence.

Synergies are not just about cooperation. Observations of relative coordination understood as metastability suggest that tendencies for competition and cooperation are present at the same time [14]. Thus the limited (and limiting) view that "competitive efforts inherently teach the values of getting more than others, beating and defeating others, seeing winning as important, and believing that opposing and obstructing the success of others is a natural way of life" ([65], p. 373), may be replaced by one that views competition and cooperation as complementary [11].

A key point of this new perspective is to establish meaningful goals that promote the emergence of synergies (relatively coordinated, metastable entities) which work in a co-ordinated and flexible manner, and take on the dimensions of a new and more complex organism capable of greater functional action diversity (see the theory of coopera-tive–competitive intelligence in more depth in [28]). This process is not only influenced by the personal (near static) characteristics of each team member: finding team solutions involves the interaction of multiple constraints nested in different level and time scales. It is necessary that the co-adaptive dynamics of the team members form self-enhancing, positive, feedback loops that accelerate the team as a system toward common converging temporary solutions. Some solutions can be temporarily stabilized (become subjectively more attractive than others) and others may lose their stability giving a way to other possibilities. It is this co-adaptive loop that gives a rise to the interplay of member' sub-jective feeling of "tension" and "letting go" in the process of co-creation [27].

Students, student teams and student–teacher teams self-organize when sharing a meaningful goal. Then, there is no need to establish a hierarchical relation between teachers and students. Both conform to a system that interacts with the environment in another scale (see Figure 2). Teachers cease to be the guiders of the process, to become also learners, which have to adjust co-adaptively to the environmental constraints im-posed to the team.

Teams are self-sustained and functional as long as the properties of individual components can be manifested while, in turn, being influenced by others in a flexible and metastable way. Interaction with varied and challenging environments determines the emergence of creativity outputs in teams. Processes of cooperation between the different intrinsic tendencies coexist with processes of competition between different ideas, solu-tions or actions performed by the different members.

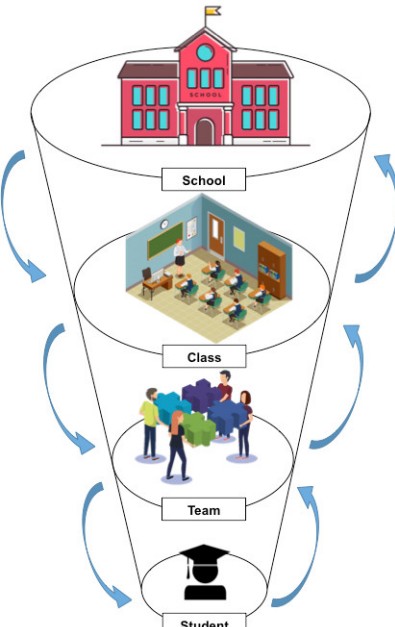

**Figure 2.** Educational levels interacting through circular or reciprocal causality. Students group with other students, students–teachers form classes, groups of classes form schools, etc. The be-havior of upper levels (e.g., the school) influences all levels down and, in turn, the interaction among individual components of lower levels (e.g., students, classes) forms the behavior of upper levels.

Different solutions can compete with each other and the organism may compete or cooperate with the environment (other teams, but also the room, the equipment, the

rules, the school or the society) depending on the type of affordances it offers. Teams can compete with other peers, but also can compete with other constraints of the environment if challenged by them. For instance, the new coronavirus disease 2019 (COVID-19) rules have prevented many games that students like to play. If teams are interested in looking for solutions to design new and funny games that respect the distance of COVID-19 barriers, the latter rules may compete with their interests and constrain their behavior, requiring the students to be cooperatively creative. Teams are pushed to decide on the basis of individual creative ideas, and other's ideas emerging while designing or practicing the new games. Some ideas will cooperate, e.g., to stay in a special location of the school and to transform it through equipment or signaling, while others will compete because they will not be able to be selected together (e.g., to re-design a school room and to go playing in a public square of the city). The multiple constraints that the students have to take into account (or influence unconsciously) mean that one of these ideas will "win", but the process of selection is characterized by metastability (different solutions may be suitable and students can jump from one to another) and fluctuations (unstable solutions, periods of instability until a solution stabilizes).

## 4. Fostering Collaborative Creativity in Learning Processes

In collaborative creativity, the new synergies also embrace variability in structure and function. Such flexibility points out some limitations to the common practice of assigning roles by teachers (or by students following some rules) and fixing sequences of actions proposed by some cooperative methodologies (i.e., jigsaw, think pair share, numbered heads together…), as those strategies can prevent the spontaneous formation of more efficient organizations. The self-organization of the team objective with the environment promotes the emergence of roles among team members, in all likelihood different to the pre-planned ideas of teachers or imposed leaders. Such imposed structures could prevent the teams from self-organizing in the most effective way according to their immediate perceived affordances.

The use of brainstorming or similar methods assumes a linear and sequential way of thinking or a linear behavior of the groups (perceive–think–act). CD changes this view, as the appearance of a new idea can emerge at any moment and change the whole process. It is also possible that the team switches between different solutions obeying a metastable dynamic. Diverse solutions, jumping from one to another or being influenced suddenly by a new idea that transforms the entire process are the norm, not the exception, of all team (and individual) creative processes. Referring to the dynamics of the creative process, Guastello [66] analyzed transcripts from three problem-solving discussions to show that productivity was chaotic over time, evidenced by a positive Lyapunov exponent of the time series. The author suggested that creative processes begin as a near random combination of ideas, which circulate through the group or culture.

This non-linear and emergent process is evidenced in improvisational settings, where the timescale of acting coincides with that of perceiving. Actions are not determined only by intentionality, but by the constant adaptation to the environment. Neuroscience shows that these processes probably occur outside conscious awareness and beyond volitional control [67]. However, neuroscience also shows that the "insights" arise mainly from non-conscious, non-reportable processes that enable problem re-structuring. The "Aha!" experience is based on the sudden emergence of insight, rather than arising as a result of linear or sequential processes that bring people progressively closer to the solution [68,69]. This would suggest that the process followed to solve problems on slower time scales than those from improvisational settings (opposition sports, conversations, art improvising, etc.) often follows also a non-linear process where the solutions emerge and are not pre-planned. Some artists use certain tricks to get distracted, to go into a non-conscious level where ideas emerge easily. On the other hand, experience and work is mandatory to improve creative behavior (as in Picasso's quote "Inspiration exists, but it has to find you working"), as personal traits and skills interact

with task demands and environmental constraints to afford the emergence of new solutions.

The complex and dynamic nature of creativity could explain the failure of some cooperative strategies to produce really creative teams, as they force students to go round and round on the same problem, preventing new perspectives from being seen, to look out of the box, and preventing the natural and spontaneous emergence of new solutions. Strategies based on a goal, a plan, the assignment of responsibilities, working through defined steps until the goal is achieved, can limit the emergence of unexpected solutions, different to any that teachers could anticipate [70]. Teachers do not know all possible solutions of a task: if the outputs are foreseeable it probably means that teams have not explored enough and have not exploited their diversity potential. Teachers can manipulate personal or environmental constraints to promote the creativity and autonomy of the students. Because of their experience, teachers can use 'tips' that allow students to discover new solutions.

## 5. Conflicts and Discussions when Innovating

Working together may involve disagreement, tension and stress, which are a reflection of different personal values among individuals who share common goals. Instead of minimizing the differences, adequate pedagogical strategies create supportive environments to share ideas and improve them through the ideas of others [70]. In diverse teams, individuals work autonomously and in cooperation with others, in a metastable mode of operation. This phenomenon is replicated at all scales, not only to coordinate individuals, but also to coordinate components and processes of the same person, or parts of those components (coordination among nerve cells, among the different organs of the digestive system or the respiratory system, etc.). Literature has shown how parts of the brain exhibit tendencies to function autonomously at the same time as they exhibit tendencies for coordinated activity [36,42,71,72]. This is possible because all the structures or parts of the same organism have a shared purpose, and they self-organize to achieve it.

Teachers and students constitute a community with a shared purpose that constrains the creative process. Hill et al. [70] argue that it is effective to follow some rules of engagement based on keeping conflicts focused on ideas rather than personalities. These rules call for mutual trust, mutual respect, and mutual influence, as well as questioning everything and seeing the whole. In fact, intellectual conflict (competition again?) is viewed as a basic ingredient for innovating. In our opinion, these rules should not be mandatory—something literally verbalized and reproducing the controlling behaviors of other approaches—but rather should emerge from the created dynamics, the created atmosphere in the community. Such an atmosphere consists of trusting the students' potentialities. Teachers can design meeting places where students feel comfortable and have the confidence to show diverse and non-orthodox behaviors, where different responses are respected and appreciated, yet at the same time reflected and discussed when needed.

In general, in stable and collaborative environments, complex living systems tend to attain low functional action entropy potentials, while in uncertain and non-cooperative environments, they enlarge the functional action entropy potential in order to satisfy the goal constraints under increasing variety of suppressing perturbations by the environment [28]. Does this mean that cooperation can be counterproductive to foster creativity? Not at all. On the contrary, the team has to cooperate to increase their functional diversity potential. The cooperation is manifested as increased shared integrative information (i.e., certainty) as seen within the team. This integrated information or functional diversity potential of the cooperative unit, i.e., the team, is manifested as functional action entropy potential, i.e., uncertainty, to the external observer. That is the meaning of the entropy-information relativity principle [28]. For example, in a team sport, the diversification of intra-team passing synergies increases the number of ways to collaboratively achieve a certain goal. Intra-team synergies increase the intra-team certainty (i.e., information) of

collaboration. By contrast, the increased number of ways of passing the ball increases the uncertainty (i.e., entropy) of the team for the external observer as to which concrete passes will be performed. The adequate manipulation of environmental and personal constraints, nested in different levels and time scales, are key aspects for designing the environments where synergic creativity can emerge. Competition with other teams or with other schools can be suitable challenges for some teams. In fact, pedagogical strategies are not good or bad in themselves but functionally and contextually adequate or non-adequate. Too demanding challenges may cause a long-term suppression of the diversity potential (e.g., due to frustration), so a co-design of challenges involving all members of the team is a recommendable strategy. The leader's role is also essential for detecting the state of the team and deciding the adequacy of the challenges.

## 6. Evaluating Cooperativity and Collaborative Creativity in Educational Settings

### 6.1. Some Measures of Cooperation

Cooperativity may be formally well represented by network (graph) models in which nodes are individuals and the edges are the interactions among them. Much of what is now known as network science, particularly social network science, has its scientific roots in the studies of Jacob Levy Moreno of interpersonal relationships and interactions [73,74]. In educational settings the research of social relationships has a rich history too (e.g., [75]). Measures used, such as social cohesion were and are useful variables for objective detection of formation and stabilization of cooperative social structures as well as determination of the role of individuals within it. For example, some group phenomena such as: chains (linearly connected individuals), islands (isolated subgroups) and circles (linearly connected individuals where the last one interacts with the first one) may be very informative about the structural properties of the group. Also, some properties of individuals such as: star (the most interactive individual) or isolates (individuals with no interaction) may be readily detected using these methods. On the other hand, the modern theory of complex networks [76], although sharing some similarities to measures used in sociometry (e.g., assortativity, clustering coefficient, etc.), also provides other measures particularly suited for studying more abundant and multilevel networks, such as degree distribution (the probability distribution of the number of connections of nodes in the network), modularity (tendency of individuals to form close groups) and hierarchical modularity (tendency of existence of modules within modules within the network). It is very important to include measures of dynamic phenomena such as percolation (when small connected subgroups of individuals transit to a fully connected cooperative group, or vice versa, for a small change of the interaction strength), which are not present in the classical sociometrical research. Depending on the problem at hand many such measures can be used in studying collaboration in creative social structures (i.e., dyads, teams, organizations, etc.).

### 6.2. Some Measures of Collaborative Creativity

Collaboration may be creative in a sense that fluent, flexible and atypical (i.e., original) functional patterns of behavior may emerge without external online control. In some sense, measures of creativity should assess the dissimilarity between the collaborative (e.g., problem-solving) patterns. Dissimilarity is a property of flexibility, diversity and originality. In other words, the larger the dissimilarity between functional patterns the larger is their flexibility, fluency and atypicality (originality). In this light similarity measures come as a natural measure of creative outcomes [24]. For example, in analyzing the structure of football games the cosine similarity measure known as Tucker's congruence was able to discern the differences of collective patterns that emerged under different task boundary conditions [77]. Similarity measures are mathematically connected to another potentially useful measure: entropy [78]. For example, if we consider correlation as a similarity measure, then entropy is the negative logarithm of the fraction

of unexplained variance. However, here we note that a thorough analysis of the content of the variables used in the investigation is needed in order to achieve meaningful entropy measures with respect to the assessment of creativity. For example, randomly walking people in a wider area would have larger entropy (i.e., uncertainty of their position) than if they were packed in a smaller area. However, from this it does not follow that the former case shows a larger creativity of the group. This was obvious in Torrents et al. [79], where the entropy measures of children's kinematic activity were not related to their exploratory behavior, as measured by a time-lagged cosine similarity measure known as dynamic overlap. This was simply because the content of the data for which the dynamic overlap was used referred to the qualitative task content of their play, whereas the accelerometer time series assessed their kinematics. A suitable measure of collaborative creativity, more linked to behavioral flexibility, may also be measured as entropy of the functional diversity of synergies [28].

It has to be emphasized that diversity includes originality, with atypicality as a special case. Functional diversity may increase as a consequence of original innovation (see Table 1). That is, when the functional behavior differs from the one that is common. However, functional diversity of the group may also be a result of learning by imitation of external models of behavior, which is not a property of creativity. Of course, any learned model of behavior by imitation may then be immersed in a different context and behavioral sequence, and hence become part of the creative whole. Functional diversity potential [28] should additionally be analyzed for its qualitative task content, in order to assess the creative aspects of the behavior.

**Table 1.** Comparison between cooperative learning (CL) and coordination dynamics (CD) to foster collaborative creativity.

| Topics | CL | CD |
|---|---|---|
| Relation among individuals | Cooperation | Synergies (cooperative–competitive processes coexist) |
| Competition | Avoided | Embraced |
| Decision-making process | Fixed and sequentially linear | Self-organized, non-linear |
| Couplings among team members | Stable | Metastable (flexible and diverse) |
| Formation of teams | Prefixed roles | Spontaneous diversification and specialization |
| Cooperation level | Among students | Multilevel: among teachers, students, students–teachers, education institutions, society, etc. |
| Evaluation of creativity | Originality and functionality of the product | Self-discovered functional diversity potential |

## 7. Conclusions

Under the framework of CD, collaborative creativity involves both cooperative and competitive processes, which instead of being conceptualized as contradictory, are seen as a complementary pair. Teams create functional solutions, increase their functional diversity, and innovate, when facing sufficiently challenging environments. Spontaneous multiscale synergies emerge when education system components, sharing values and goals, interact with a competitive environment. Creativity is a consequence of such interaction that challenges the system's cooperative–competitive intelligence. As outcomes are context-dependent and complex living systems tend to produce sub-optimal behaviors, if environmental challenges are too easy or too repetitive, i.e., do not require inno-

vation, teams do not further develop their functional diversity potential. Thus, collaborative creativity in education requires sharing goals and the exposure to sufficiently new and adequately challenging environments. Only when the context demands more diversity do teams create new synergies and increase their functional diversity potential. The adequate manipulation of environmental and personal constraints and the knowledge of their critical points will prove to be key aspects in order to develop synergic creativity and address the sustainable challenges of our society.

Empirical research is warranted to explore the issues proposed in this paper and contribute to finding diverse and original collaborative solutions to address the sustainability challenges of our society in educational settings. An exciting area of research stems from the metastable mode of thinking championed by CD. There, creativity is not restricted to exclusive either–or categories and categorization but embraces the inclusive middle. As restrictive old dualisms and either–or thinking are seen to be mere limits, barriers may disappear allowing an appreciation of the full range of experience where creativity resides.

**Author Contributions:** Conceptualization, C.T. and N.B.; writing—original draft preparation, C.T., N.B., R.H. and J.A.S.K.; writing—review and editing, all authors; visualization, M.A.; funding acquisition, C.T., N.B. and M.A. All authors have read and agreed to the published version of the manuscript.

**Funding:** This work was supported by the National Institute of Physical Education of Catalonia (INEFC), Generalitat de Catalunya. M.A. is supported by the project "Towards an embodied and transdisciplinary education" granted by the Ministerio de Educación y Formación Profesional of the Spanish government (FPU19/05693). J.A.S.K. is supported by NIMH Grant MH MH080838, the Davimos Family Endowment for Excellence in Science, and the Florida Atlantic University Foundation.

**Institutional Review Board Statement:** We have not made any empirical study.

**Informed Consent Statement:** Not applicable.

**Data Availability Statement:** Not applicable.

**Acknowledgments:** The assistance provided by Sergi Garcia-Retortillo was greatly appreciated.

**Conflicts of Interest:** The authors declare no conflict of interest.

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
