# Peer review of "Metastable Coordination Dynamics of Collaborative Creativity in Educational Settings"

_sustainability, doi:10.3390/su13052696_

Round 1

Reviewer 1 Report

It is a well-written manuscript that offers a well-founded and argued theoretical framework.
This work offers a very important vision of Coordination Dynamics, offering very important results on its influence on the development of creativity.
I understand that it is a very interesting work only some bibliographic references should be reviewed. 

Author Response

Thank you very much for your review and nice comments. The bibliographic references have been revised.

Reviewer 2 Report

Dear Authors

First I would like to wish you all well

The paper entitled "Metastable Coordination Dynamics of Collaborative Creativity in Educational Settings" is an extensive and appropriate theoretical research. It uses very technical language and gets to transfer knowledge about collaborative creativity in education. I admit that sometimes it was difficult for me to understand the ideas you wanted to communicate, but I really enjoyed reading your work. 

However, I have doubts about the type of article presented. It is a Position Papper.To the best of my knowledge these investigations deal with a relevant and topical issue (you do) and confront two positions (in this sense I don't quite capture it clearly). 

I have the feeling that the work is limited to expose different variables that affect collaborative creativity, but I do not see the contrast with cooperative work.

In any case, the theoretical work done is very good, of an excellent quality, and it brings an undoubtedly novel topic to the field of cooperative learning, which has been somewhat stagnant in recent years.

The mastery of the metastable Coordination Dynamics is evident. However, I have the feeling that you constantly make statements that are not quoted. As this is a new topic, readers would appreciate knowing the primary sources. 

Therefore, I only consider that you should clarify which are the opposing positions explained in your work, improve the argumentation of some parts of the text with appropriate citations and I have suggested your acceptance for publication.

I add below a series of considerations with the aim of improving your excellent work:

The overall introduction is exciting and attention-grabbing. However, I find that there are too many unquoted statements. I believe in the novelty of your work, but I need to know the primary sources.

Line 37. “Creativity is essential to deal with the challenges of sustainable development”. Please quote this statement. UNESCO, in its Education for Sustainability document, defends this position.

Line 43-48. Maybe it might be required quote same of this statement.

Lines 48 – 72- Again, I consider that they make an introduction with categorical and very interesting statements, but without adequate bibliographical justification. Please reference as far as possible.

In describing the variables that condition cooperative behavior, gender is described as one of them. Please review these papers and include them as references if you consider them appropriate:

The overall introduction is exciting and attention-grabbing. However, I find that there are too many unquoted statements. I believe in the novelty of your work, but I need to know the primary sources.

Line 37. “Creativity is essential to deal with the challenges of sustainable development”. Please quote this statement. UNESCO, in its Education for Sustainability document, defends this position: https://en.unesco.org/themes/education-sustainable-development

Line 43-48. Maybe it might be required quote same of this statement.

Lines 48 – 72- Again, I consider that they make an introduction with categorical and very interesting statements, but without adequate bibliographical justification. Please reference as far as possible.

In describing the variables that condition cooperative behavior, gender is described as one of them. Please review these papers and include them as references if you consider them appropriate:

Baena-Morales, S., Jerez-Mayorga, D., Fernández-González, F. T., & López-Morales, J. (2020). The Use of a Cooperative-Learning Activity with University Students: A Gender Experience. Sustainability, 12(21), 9292. [doi:10.3390/su12219292]

Rodger, S. & Murray, H.G. & Cummings, A.L.. (2007). Gender differences in cooperative learning with university students. Alberta Journal of Educational Research. 53. 157-173.

150-156 I do not understand why they do not quote these phrases, they are the scientific arguments to justify the quality of their work.

Sometimes I have the feeling of getting a little lost in the reading of the text. This kind of articles are not very common, but I think they should present in a clearer way the two main opposing ideas. Is this possible?

I would like to see examples or an apparatus of practical applications. There is table 1 but I think that some more practical examples may help the future reader to understand the differences between CL and CD.

Conclusion

I have the feeling that the conclusion could be a bit broader. You have developed an excellent theoretical work, do you consider the length of your conclusion sufficient?

All in all, I would like to congratulate you once again for your excellent work.

Author Response

Dear reviewer, thank you very much for your interesting comments. We are all fine and hope that you too.

We agree that we have not been precise when describing this paper as a position one. This paper aims showing how collaborative creativity can be enhanced, embracing the advances offered by Cooperative Learning and Collaborative learning, but considering the principles that Coordination Dynamics offers to understand any human coordinative behaviour. It is not our aim to confront two positions, and for this reason we have removed the term “position paper” and added a sentence about this at the end. Cooperative strategies are welcomed, but also competitive ones, and the use of any other strategical pedagogies when the context requires it.

We answer to your detailed comments:

- Line 37. “Creativity is essential to deal with the challenges of sustainable development”. Please quote this statement. UNESCO, in its Education for Sustainability document, defends this position.

Answer: Thank you very much for this suggestion. We have added it.

- Line 43-48. Maybe it might be required quote same of this statement.

Answer: Sorry, but we have not found the place to introduce that quote.

- Lines 48 – 72- Again, I consider that they make an introduction with categorical and very interesting statements, but without adequate bibliographical justification. Please reference as far as possible.

Answer: We added new references.

- In describing the variables that condition cooperative behavior, gender is described as one of them. Please review these papers and include them as references if you consider them appropriate:

Baena-Morales, S., Jerez-Mayorga, D., Fernández-González, F. T., & López-Morales, J. (2020). The Use of a Cooperative-Learning Activity with University Students: A Gender Experience. Sustainability, 12(21), 9292. [doi:10.3390/su12219292]

Rodger, S. & Murray, H.G. & Cummings, A.L.. (2007). Gender differences in cooperative learning with university students. Alberta Journal of Educational Research. 53. 157-173.

Answer: Thank you, we have added these references and the necessity to take into account the gender perspective.

- 150-156 I do not understand why they do not quote these phrases, they are the scientific arguments to justify the quality of their work.

Answer: We have tried to improve the references in the introduction.

- Sometimes I have the feeling of getting a little lost in the reading of the text. This kind of articles are not very common, but I think they should present in a clearer way the two main opposing ideas. Is this possible?

Answer: As stated before, the aim was not to confront opposite ideas, we have added a sentence at the end to clarify this point. 

- Conclusion

I have the feeling that the conclusion could be a bit broader. You have developed an excellent theoretical work, do you consider the length of your conclusion sufficient?

All in all, I would like to congratulate you once again for your excellent work.

Answer: We have tried to improve the conclusions and applications of the paper. Thank you again for your nice comments and suggestions.

Reviewer 3 Report

Some recommended observations that will enhance the quality of the article:

The theoretical part of the article must be supplemented with essential information about Sustainability as a Goal, as this is the authors’ key aim and context. This is also important in terms of the philosophy and content of the journal. This place should be one of the main and most broadly substantiated places in the theoretical analysis.

In general, the theoretical analysis is presented in a rather complex style. There are no clearly named techniques and/or principles of theoretical analysis. Thus, it is not clear how scientific literature was analysed.

Some concepts require more precise clarity; for example, how Cooperative Learning is related to Collaborative Creativity. Do the authors of the article treat the concepts-processes “Cooperative Learning” and “Collaborative Learning” as identical concepts and processes? It is necessary to explain these concepts clearer.

The conclusions could be more specifically formulated and need to be linked to the contexts of sustainability. The use of quotations in conclusions (line 535; 539) is not recommended. If these quotations and references are unavoidable and necessary, then it should be recommended to think about the part of the discussion.

The presented research limitations and possible future perspectives would strengthen the content and value of the article.

Author Response

Thank you very much for your nice comments and suggestions. We have introduced more references to Sustainability as a Goal in the introduction of the paper.

About the terms Cooperative Learning, Collaborative Creativity and Collaborative Learning, we agree with you that we had not clarified enough these terms. For this reason, the following paragraph has been added:

Both terms, Cooperative Learning (CL) and collaborative learning are used to define processes that increase sustainable development competencies [6]. Although the first one is more common in the education literature, the second one also refers to instructional arrangements that involve two or more students working together on a shared learning goal. It is sometimes differentiated from CL because it does not specify the division of labor [7]. In the creativity literature, it is more frequent to use the term Collaborative Creativity, that refers to processes of creation based on individuals working together [8].

About the conclusions, we added references to the context of sustainability and removed the quotations, as they are presented before.

We have also added a final paragraph related with the limitations and future perspectives. Thank you for your useful suggestions.